# Multiple Modulation of Acid-Sensing Ion Channel 1a by the Alkaloid Daurisoline

**DOI:** 10.3390/biom9080336

**Published:** 2019-08-02

**Authors:** Dmitry I. Osmakov, Sergey G. Koshelev, Ekaterina N. Lyukmanova, Mikhail A. Shulepko, Yaroslav A. Andreev, Peter Illes, Sergey A. Kozlov

**Affiliations:** 1Shemyakin-Ovchinnikov Institute of Bioorganic Chemistry, Russian Academy of Sciences, ul. Miklukho-Maklaya 16/10, 117997 Moscow, Russia; 2Institute of Molecular Medicine, Sechenov First Moscow State Medical University, Trubetskaya str. 8, bld. 2, 119991 Moscow, Russia; 3Rudolf-Boehm-Institut für Pharmakologie und Toxikologie, University of Leipzig, 04107 Leipzig, Germany

**Keywords:** daurisoline, ASIC1a channels, potentiator, channel desensitization

## Abstract

Acid-sensing ion channels (ASICs) are proton-gated sodium-selective channels that are expressed in the peripheral and central nervous systems. ASIC1a is one of the most intensively studied isoforms due to its importance and wide representation in organisms, but it is still largely unexplored as a target for therapy. In this study, we demonstrated response of the ASIC1a to acidification in the presence of the daurisoline (DAU) ligand. DAU alone did not activate the channel, but in combination with protons, it produced the second peak component of the ASIC1a current. This second peak differs from the sustained component (which is induced by RF-amide peptides), as the second (DAU-induced) peak is completely desensitized, with the same kinetics as the main peak. The co-application of DAU and mambalgin-2 indicated that their binding sites do not overlap. Additionally, we found an asymmetry in the pH activation curve of the channel, which was well-described by a mathematical model based on the multiplied probabilities of protons binding with a pool of high-cooperative sites and a single proton binding with a non-cooperative site. In this model, DAU targeted the pool of high-cooperative sites and, when applied with protons, acted as an inhibitor of ASIC1a activation. Moreover, DAU’s occupation of the same binding site most probably reverses the channel from steady-state desensitization in the pH 6.9–7.3 range. DAU features disclose new opportunities in studies of ASIC structure and function.

## 1. Introduction

Acid-sensing ion channels (ASICs) are proton-gated members of the degenerin/epithelial Na^+^-channel superfamily. To date, four genes (ACCN1–ACCN4)—encoding six subunits that can form homo- or heterotrimeric channels—have been found and cloned [1,2,3]. ASICs, especially the ASIC1a isoform, are widely expressed in the peripheral and central nervous systems, where they are involved in physiological and pathological processes such as synaptic plasticity, neuronal injury, and nociception [4,5,6]. ASIC1a activation mediates neurodegeneration and death under pathological conditions such as cerebral ischemia, inflammation, and traumatic injury [7,8,9,10]. Because ASIC1 and ASIC3 also contribute to hyperalgesia, the molecules that inhibit ASIC1 and ASIC3 activity have been recognized as prospective analgesic drugs [11,12,13,14,15,16].

For the central nervous system, the major impact of proton sensing is attributed to ASIC1a containing homotrimeric or heterotrimeric ion channels. This subtype differs from others in terms of biophysical properties, pH sensitivity, and ligand specificity. ASIC1a usually produces a transient current in response to fast extracellular acidification. But ASICs can reach steady-state desensitization (SSD) that means they become desensitized when exposed to a moderate reduction in extracellular pH (which itself is not sufficient for robust activation) and do not respond to further drops in extracellular pH [17,18].

The main developments in investigations of ASIC1 have related to the use of specific ligands: Peptide toxins from natural venoms, neuropeptides, and small organic molecules. These ligands have different pharmacological effects on the channel. At nanomolar concentrations, the spider peptide toxin PcTx1 and the snake peptide toxins mambalgins inhibit mammalian ASIC1 currents by stabilizing the channel’s inactive state and its closed state, respectively [19,20]. The coral snake toxin MitTx—which consists of a heteromeric complex of Kunitz- and phospholipase-A2-like proteins—preferably activates ASIC1 channels [21]. All these toxins have been shown to bind near the acidic pocket that is located in the extracellular domain between adjacent subunits [20,22,23,24]. At millimolar concentrations, the synthetic molecule 2-guanidine-4-methylquinazoline activates ASIC3 at pH 7.4, whereas it inhibits other ASIC isoforms by changing the pH dependence of both activation and SSD [25,26]. At micromolar concentrations, mammalian RF-amide neuropeptides slow the kinetics of desensitization, induce a sustained current, and inhibit the SSD of ASIC1a [27,28,29]. Other neuropeptides—big dynorphin and dynorphin A—decrease ASIC1a’s proton sensitivity at SSD [30].

Here, we demonstrated a novel mode of action for the natural, low-molecular-weight compound daurisoline (DAU; PubChem CID 51106), exerted on rat ASIC1a expressed in oocytes. DAU is distributed in a variety of Chinese medicinal herbs such as *Menispermum dauricum*. DAU is commercially available, and has been reported to have neuron-protective and anti-inflammatory properties, and can effectively block autophagy [31,32,33]. It has been shown that DAU causes muscle relaxation inhibiting L-type calcium channels currents [34] and possesses antiarrhythmic properties inhibiting hERG current in a voltage-dependent manner at micromolar concentrations [35]. We found that DAU potentiated the integrated response of the channel to acidosis and inhibited both the SSD and the activation of ASIC1a by moderate acidification.

## 2. Materials and Methods

### 2.1. Ethics Statement

This study was carried out in strict accordance with the World Health Organization’s International Guiding Principles for Biomedical Research Involving Animals. The protocol was approved by the Institutional Policy on the Use of Laboratory Animals of the Shemyakin-Ovchinnikov Institute of Bioorganic Chemistry RAS (Protocol Number: 251/2018 26.02.18).

### 2.2. Expression in Xenopus laevis Oocytes

Unfertilized oocytes were harvested from female *Xenopus laevis*. All procedures were performed in agreement with the guidelines of ARRIVE (Animal Research: Reporting of In Vivo Experiments) and the “European convention for the protection of vertebrate animals used for experimental and other scientific purposes” (Strasbourg, 18.III.1986). Female frogs, maintained at 20 ± 2 °C on 10 h light/14 h dark cycle, were anaesthetized in a 0.17% solution of procaine methanesulphonate (MS222), and then a small part of the ovary was removed through a small incision on the abdomen. The incision was sutured, and the animal was held in a separate tank until it had fully recovered from the anesthesia. The animals did not show any signs of postoperational distress. For each animal, the interval between surgeries was at least three months. The follicle membranes were removed from the oocytes by treatment with 1 mg/mL of collagenase in ND96 medium (96 mM NaCl, 2 mM KCl, 1 mM MgCl_2,_ and 5 mM HEPES titrated to pH 7.4 with NaOH) lacking calcium for 2–3 h at room temperature. Healthy stage IV and V oocytes were sorted and stored in ND96 medium solution. Then, 16 to 18 h after isolation, oocytes were injected with 2.5–10 ng of cRNA using the Nanoliter 2000 microinjection system (World Precision Instruments, Sarasota, FL, USA). This cRNA was synthesized from PCi plasmids containing the rat ASIC1a isoform. We confirmed the integrity of the ASIC gene through DNA sequencing of all the inserted fragments. After injection, the oocytes were kept for 2–3 days at 18 °C and then for up to 7 days at 15 °C in ND-96 medium supplemented with gentamycin (50 μg/mL).

### 2.3. Electrophysiological Experiments

Two-electrode voltage-clamp recordings on oocytes were performed using the GeneClamp500 amplifier (Axon Instruments, Inverurie, UK) at a holding potential of −50 mV. The data were filtered at 20 Hz and digitized at 100 Hz using the L780 AD converter (L-Card, Moscow, Russia). Microelectrodes were filled with 3 M KCl. The external solution was ND-96 with pH adjusted to 7.8, 7.4, 7.2, 7.0, 6.8, or 6.6. Proton-activated currents through ASIC channels were elicited by application of ND-96, in which 5 mM of HEPES was substituted with 10 mM MES for solutions’ pH < 7.0. The stream flow about 1 mL/min and a solution exchange rate about 1 mL/s were achieved in the recording chamber using a computer-controlled valve system.

### 2.4. Statistical Analysis

The electrophysiological data were analyzed using OriginPro 8.6 software (OriginLab, Northampton, MA, USA). Curves were fitted using the logistic equations: a) F_1_(x) = A − A/(1 + (x/x0)^nH^), where F(x) is the current amplitude at DAU concentration x, A is the maximal current amplitude, x0 is the EC_50_ value, and n_H_ is the Hill coefficient (slope factor); b) F_2_(x) = A/((1 + (x/[pH_50_1])^nH1^)*(1 + (x/[pH_50_2])^nH2^), multiplication of two logistic equations where [pH_50_1] is the half-maximal concentration of protons binding to pool of high-cooperative sites (or 0.5 probability of “high-cooperative” sites’ occupancy) and [pH_50_2] is the half-maximal concentration of protons binding to the non-cooperative site (or 0.5 probability of “non-cooperative” site occupancy); n_H_1 and n_H_2 are the Hill coefficients for the first and second logistic equations, respectively; A is the maximal current amplitude; c) F_3_(x) = ((a1 − a2)/(1 + (x/x0)^nH^)) + a2, where F(x) is the response value at a given peptide mambalgin-2 concentration; x is the concentration of the peptide; a1 is the control response value (fixed at 100%); a2 is the response value at maximal inhibition (% of control); x0 is the IC_50_ value; and n_H_ is the Hill coefficient. The maximum amplitude (I_max_) was calculated for each oocyte by individual fitting, and the data were normalized to it, then the normalized data were averaged and fitted by the logistic equation F_1_(x).

The rate of curves decay was fitted using a single exponential equation, F_4_(x) = A1*e^(−x/τd)^ + A0, where τ_d_ is the τ of desensitization.

All data are presented as mean ± SEM. The significance of the data differences was determined by a Tukey’s test ANOVA, with the significance level * *P* < 0.05. For normalized data difference significance of the non-parametric Kruskal–Wallis ANOVA test was utilized with the significance level * *P* < 0.05. In all experiments the responses were recorded by a person not blind to the treatment.

### 2.5. Mathematical Modeling of ASIC1a Activation

Kinetic models used to simulate ASIC1a activation were based on the conventional rate theory and used as independent forward and reverse rate constants to simultaneously solve first-order differential equations representing the transitions between all possible states of the channel. Differential equations were solved numerically by using the algorithm analogous to that described by [36]. The maximum values of the obtained current amplitudes were fitted using the logistic equation F_2_(x) and analyzed by OriginPro 8.6 software (OriginLab, Northampton, MA, USA).

### 2.6. Chemical Reagents and ASIC1a Ligands

DAU and salts for a buffers’ preparation were obtained from Sigma-Aldrich (Steinheim, Germany). Lindoldhamine was obtained using isolation procedure described by [37]. Recombinant analogue of mambalgin-2 and the solid-phase synthesized FRRF-NH_2_ peptide were obtained as a gift from collaborators at the Shemyakin–Ovchinnikov institute in Moscow. Fresh working solution of the ligands in the ND96 buffer was prepared immediately before the experiments.

## 3. Results

### 3.1. DAU Promotes the Second Peak Component of the ASIC1a Current

The ASIC1a channel in the presence of DAU demonstrated a two-component current in response to acidification (Figure 1). This is a very unusual effect on ASIC1a’s response to a fast acidic stimulus. The unusual additional current showed the same kinetics of desensitization as the normal acid-induced current.

At the same time, DAU pre-incubation produced no effect on the first (“normal”) peak component of current, which was generated by a pH drop from 7.4 or 7.8 to 5.5. The previously described ASIC ligand lindoldhamine [37] with a very close structure to DAU, did not have this effect (data not shown). The second peak component’s growth depended on the concentration of DAU, which we applied in the range of 0.01–0.3 mM (Figure 2a); fitting the second peak’s amplitude data with the logistic equation resulted in a half-maximal concentration (EC_50_) value of 23 ± 3 μM and a Hill coefficient (n_H_) of 1.8 ± 0.4 (Figure 2b). It should be noted that the rate of desensitization for the second component was the same as that for the (control) main peak, as the τ of desensitization constants did not differ significantly (5.35 ± 1.15 s for control vs. 4.95 ± 1.39 s for the second peak; n = 7).

### 3.2. Comparison of DAU and RF-Amide Peptide Effect on ASIC1a Current Desensitization

The ASIC1a isoform’s ability to produce a non-uniform response to proton activation in the presence of some modulators is known, as is the fact that mutated channels can gain such properties as well. The most prominent effect for wild-type channels can be observed during the application of RF-amide peptides, which induce a slowly decreasing, sustained current component of the ASIC1a response [28]. To compare the kinetics of the second (DAU-induced) component and the sustained (RF-amide-peptide-induced) component, we applied DAU and the FRRF-NH_2_ peptide to the same oocyte-expressed ASIC1a channels (Figure 3). DAU and the peptide affected the channel’s desensitization process differently. The peptide inhibited the desensitization of the main peak; this was further expressed in the development of a sustained, non-desensitized component of the current, which lasted as long as the pH stimulus continued (Figure 3a, blue trace). On the other hand, DAU also exhibited an apparent second component, which began to desensitize before the end of the stimulus (Figure 3a, red trace).

To quantify and compare the effects that DAU and the peptide had on the channel’s desensitization, we chose the parameter τ_dec_ (the amount of time that the amplitude of the current takes to decay to e times its initial amplitude). The calculated τ_dec_ value for the proton-induced main peak did not significantly change in the presence of DAU (2.41 ± 0.15 s for control, n = 12; 2.3 ± 0.1 s for DAU, n = 10). The second DAU-induced peak had τ_dec_ = 2.5 ± 0.4 s (n = 6), which also did not differ statistically from the control (Figure 3b). In contrast, the peptide induced a decrease in the main component’s desensitization, τ_dec_ = 4.15 ± 0.55 s (n = 5; Figure 3b). Thus, DAU has very little effect on the desensitization of the main peak (and even slightly accelerates it), but the RF-amide-peptide significantly slows this process down.

### 3.3. DAU-Induced Recovery of Desensitized ASIC1a Current

Acid-induced SSD is a distinguishing characteristic of ASICs, and some ligands can shift the pH-dependence of this process. DAU can also affect SSD. The pre-application of DAU led to the full recovery of the currents (Figure 4a). This effect depended on the DAU concentration; the amplitude of the current fit with the logistic equation in the DAU concentration range of 0.01–3 mM, resulted in an EC_50_ value of 137.8 ± 27.6 μM and a n_H_ value of 0.82 ± 0.05 (Figure 4b). The maximal recovery effect was achieved at 1 mM concentration, and the 3 mM concentration showed a stable saturation effect with a tiny decrease.

The conditioning pH value, which causes 50% desensitization of the current (pHSSD_50_) was shifted to a more acidic area in the presence of DAU (from 7.27 ± 0.01 for the control to 7.07 ± 0.03 for 1 mM DAU; Figure 4c). This shift was accompanied by a significant decrease in the Hill coefficient for the protons (n_H_ = 13.3 ± 1.2 for the control vs. n_H_ = 3.81 ± 0.75 for 1 mM DAU), thus highlighting that ASIC1a has a different conformational state in the presence of DAU.

The development of the second peak at conditioning pH 7.0 was normal for DAU-treated channels, but the acidification of the external solution decreased the effective concentration of DAU. The amplitude data for the second component fit with the logistic equation, showed a 40-fold reduction of DAU’s apparent affinity at pH 7.0 (EC_50_ = 913 ± 50 μM, n_H_ = 2.01 ± 0.18; Figure 5).

The presence of more acidic media decreased the τ_dec_ value to 1.8 ± 0.1 s for the recovered main peak and 1.67 ± 0.11 s for the second peak (n = 5). However, the ratio of the τ_dec_ values between the main and second components was the same as at pH 7.8.

### 3.4. DAU’s Effect on the Proton Dependence of ASIC1a Activation

For ASIC1a activation by protons alone, we more precisely registered the channels’ response as being in the pH 7.0–4.0 range. Surprisingly, the amplitudes that we obtained on channels expressed in *X. laevis* oocytes had an unsatisfactory fit with the usual logistic equation F_1_ (see details in Material and Methods) (Figure 6a, dotted line). In the range of pH from 7 to 4, the channel’s activating dependence graph did not suit well an ordinary sigmoid; rather, it was asymmetric (Figure 6a, solid line). The same results were obtained on the rat ASIC1a channels expressed in CHO cells (see data replotted from Figure 1k in [38]; Figure 6b). The curve is steep in the narrow stimulation area (up to pH 6.5) and flattens out in an area of stronger acidification.

The channel-activation process could be considered as a result of the simultaneous protonation of a pool of “high-cooperative” sites and an additional “non-cooperative” site(-es) of the channel. The dose dependence of the channel activation in this case could be described by the equation F_2_ (see details in Materials and Methods). The experimental data fitted well with this equation indicating the cooperation of unequal proton sites during the channel activation (Figure 6a,b,d, black solid line). The calculated values for the pH_50_1 and pH_50_2 were 6.67 ± 0.01 and 6.59 ± 0.05, respectively, for oocytes, and 6.7 ± 0.13 and 6.45 ± 0.12, respectively, for CHO; n_H_1 and n_H_2 were 6.75 ± 0.54 and 0.96 ± 0.06, respectively, for oocytes, and 2.1 ± 0.2 and 0.99 ± 0.29, respectively, for CHO (Figure 6c).

The co-application of 1 mM DAU with an acidic stimulus changed the proton dependence of the ASIC1a activation (Figure 6d, red line). The amplitude’s fit with equation F_2_ in the pH range of 7 to 5 indicates that DAU appreciably shifted the pH_50_1 towards a more acidic value (from 6.67 ± 0.01 for the control vs. 6.2 ± 0.01 for DAU; n = 7, *P* < 0.05) and that it did not reliably affect the pH_50_2 value (6.59 ± 0.05 for the control vs. 6.48 ± 0.06 for DAU; n = 7, *P* = 0.07). The Hill coefficient n_H_1 showed a tendency to change in the presence of DAU, although this change did not reach the level of statistical significance (from 6.7 ± 0.5 for the control to 6.36 ± 0.77 for 1 mM DAU; n = 7). This result suggests that DAU changed the channel activation by interfering with the high-cooperative proton sites but that it did not influence the non-cooperative site.

### 3.5. Competition of DAU with Mambalgin-2

Three-finger toxins mambalgin-1 and 2 (Mamb-1 and 2) isolated from black mamba venom for which the binding site on chicken ASIC1 was determined by cryo-electron microscopy have an interesting inhibitory effect [20,24]. DAU has an opposite potentiating effect on ASIC1a, ultimately leading to greater conduction of ions through the membrane due to the second peak. At a concentration of 1 μM, we showed that Mamb-2 strongly inhibited the current (83 ± 2.7%) when the pH dropped from 7.8 to 5.5 pH (Figure 7a,b). In the presence of 300 μM DAU, the inhibitory effect of toxin (1 μM) was significantly reduced to 39.2 ± 6.3% (n = 5; Figure 7a,b). In order to check if Mamb-2’s and DAU’s binding sites on the receptor overlap we measured the dose–response inhibitory effect of Mamb-2, both alone and in the presence of 300 μM DAU (Figure 7c). Both compounds were applied in mixtures for 15 s before activating ASIC1a with protons. The amplitude fits with the logistic equation in the Mamb-2 concentration range of 0.01–3 μM resulted in IC_50_ values of 0.15 ± 0.06 μM (n_H_ = 0.82 ± 0.05) for Mamb-2 alone and 0.14 ± 0.02 μM (n_H_ = 1.15 ± 0.12) for Mamb-2 with DAU (n = 5).

Because these IC_50_ values were not significantly different, there is no competition between Mamb-2 and DAU for an identical site or a partial overlapping of their binding sites. The saturation of Mamb-2’s inhibitory effect at 73.3 ± 7.8% in the presence of DAU also proves that there is no competition between these ligands. Thus, we assume that DAU allosterically interferes with the toxin’s effect and that Mamb-2’s and DAU’s binding sites do not overlap.

### 3.6. ASIC1a Activation Model

The behavior of the channel interacting with protons could be described by a relatively simple model (Model 1 in Appendix A). This model includes several closed states, which, with probability α or γ, can eventually become open or desensitized, respectively. However, this model does not produce the correct dose dependence like the experimental one (see Appendix A). As a result, we proposed the following model 3–10 (Scheme 1):

According to this model: a) The channel has 9 high-cooperative sites and 1 additional site for protons binding; b) this additional site is characterized as independent, non-cooperative, and less affine; c) the binding and dissociation of each subsequent proton to the channel is attenuated by the factors p and q for high-cooperative, r and s for non-cooperative site; d) to maintain the cyclic equilibrium condition, the value p*k_on_ = r*k_on_, and q*k_off_ = s*k_off_; e) the constant of binding with the additional site (k’_on_) is equal to k_on_= 2 × 10^9^ M^−1^s^−1^, f) the dissociation constant k_off_ = 10^3^ s^−1^, and k’_off_ is 1.2 × 10^3^ s^−1^.

The model 3–10 simulation of the channel opening was carried out from conditioning pH 7.4 using pH drop stimuli in the 7.1–4.5 range. The variation of changing factors allowed us to achieve accurate simulation of data experimentally obtained on oocytes and CHO cells (Figure 6a–c red solid line). In case of oocytes the model gave an exact simulation with values of p = r = 0.8 and q = s = 0.65 (Figure 6a,c); for CHO data the exact coincidence is achieved with p = r = 0.79, and q = s = 0.6 (Figure 6b,c). Thus, the model with 10 proton binding sites allows to accurately describe the channel behavior during activation, regardless of the expression system.

## 4. Discussion

The ASIC1a channel is one of the most intensively studied ASIC due to participation in a variety of important biological processes and wide representation in organisms. In this regard, most ligands of ASICs are studied using the ASIC1a isoform [39,40]. The ligands that have been discovered so far usually have an inhibitory effect on the acid-induced current of ASIC1a by shifting the activation curve towards more acidic values or/and by shifting the SSD curve to more alkaline values [19,26,41].

Like other inhibitors of this channel, DAU shifts the pH dependence of activation towards more acidic values (Figure 6b), but this effect is combined with a shift in the pH dependence of SSD towards more acidic values. This SSD inhibition is realized via a dramatic decrease in the SSD curve’s slope, thus enhancing the channels’ excitability in a small acidic range, pH 6.8 to 7.2 (Figure 4c). A similar effect has been observed in the case of 2-guanidine-4-methylquinazoline (GMQ), which shifts the activation curve to a more acidic region and the SSD curve in the same direction, although without changing the slope [26]. Other well-studied ASIC1a inhibitors can only shift the pH dependence of SSD (in the case of PcTx1) or the pH dependence of activation (in the case of mambalgins) [41]. The conclusion is thus that, of the ASIC1a ligands studied, DAU possesses a unique combination of these two effects. It is worth noting that effect of DAU onto another known molecular target, hERG channel, is not simple too [35]. DAU shifts the steady-state inactivation of hERG as in the case of ASIC1a. On the other hand, DAU does not affect the activation gating of the hERG channel but it shifts the activation curve of ASIC1a. Finally, DAU produces the reduction of total hERG current via acceleration of the onset of inactivation, while ASIC1a current is enlarged due to the second fast-inactivated component.

The appearance of a second peak component is the most prominent effect of DAU. This effect most likely correlates with the molecule’s hydrophobic pattern because a structurally related molecule, lindoldhamine (truncated by three methyl groups, PubChem CID 10370752) did not produce second peak component. The ability of ASIC1a to exhibit a two-component current in response to acid stimulation has been shown for some endogenous peptides. One such example relates to RF-amide peptides, which induce a long-lasting (sustained) current component [29]. Our comparative analysis revealed quantitative and qualitative differences between the second (DAU-induced) component and the sustained component that appears in the presence of peptide. The sustained (RF-amide-peptide-induced) component lasts as long as the acid stimulus continues, and the desensitization time of the main peak increases by 1.5 times. DAU caused the second peak that has the same desensitization time as the main current, and it may indicate that these ligands have different binding sites.

Another example of a similar sustained component emergence was observed after the replacement of acid residues in the palm domain with the corresponding amides, which supposedly simulated protonation in this area [42]. These mutations led to the appearance of various sustained currents in response to acid stimulation in the mutated channel. In another channel-mutagenesis study, the palm domain was shown to be essential to the recognition of RF-amide peptides and to interfere with the conformational changes that are necessary for inactivation [43]. Thus, RF-amide peptide’s binding site may be located in the area surrounding the palm domain. Taking into account the differences in the effects discussed above, we presume that DAU’s binding site is located outside the palm domain.

Some indirect data about possible binding sites for DAU can be extracted from the competition of DAU with the Mamb-2 toxin. The presence of DAU significantly reduced Mamb-2’s inhibitory effect on ASIC1a (Figure 7a), but the toxin’s affinity to the channel did not change (Figure 7b). This observed reduction of the inhibitory effect resulted from the presence of a permanent DAU-generated current (equal to 20% of the total current) that was not sensitive to Mamb-2. Thus, the binding sites for DAU and Mamb-2 did not overlap. We observed that DAU rather allosterically interfered with Mamb-2’s inhibitory effect and the mechanisms of their effects on the channel completely overlapped. According to the latest cryo-EM data, the extracellular thumb domain, rather than the acidic pocket, is the binding site for the toxin [24]. Once we excluded the palm and thumb domains from the pool of high-cooperative sites, the acidic pocket could be the most likely candidate for the DAU binding site.

The proton-binding sites had an unequal impact on the pH dependence of channel activation. The expanded set of experimental points did not satisfactorily fit with the typical logistic equation. We assumed that the activation of the channel is a superposition of two simultaneous events: Protons binding in both the pool of high-cooperative sites and in one non-cooperative site; we thus derived equation F_2_ for this model. It is important to note that the pH-response curves as measured in oocytes (Figure 6a; present study) and CHO cells (Figure 6b; re-plotted from [38]), both transfected with rat ASIC1a, were quite similar. That proves that the deviation of the curve from a perfect sigmoidal form is not due to some different properties of the expression systems used. Noteworthy that a similar pH-dependence in mouse ASIC1a activation was described [28] but was not properly interpreted.

By fitting the pH dependence of the channel activation in the presence of DAU, we showed that the parameter pH_50_1 changed significantly but that the parameter pH_50_2 did not. This may indicate that DAU binds at some high-cooperative sites or that it interferes with processes occurring after the protonation of these sites. 

The extensive experimental data on ASIC ligands and the novel data obtained in this study confirm that fine-tuning the channel’s functioning occurs through a combination of the activation and SSD processes [26,41,42]. The endogenous ligands’ SSD modulation [29,37,44] points out the physiological importance of SSD regulation for neurons that express ASICs [13,30]. The current, after having been desensitized at the conditioning pH (7.0), recovered almost completely in the presence of DAU. Consequently, the pHSSD_50_ shifted to a more acidic area, and the channel needed more acidic conditions for complete desensitization. The decrease in the slope of the SSD curve could be a consequence of DAU binding to the pool of high-cooperative proton sites. DAU binding to this pool may interrupt the protons’ cooperative action on desensitization, which results in a 3.5-fold decrease in the Hill coefficient. The competition between protons and DAU also can be seen in Figure 5, in which an increase in acidification from 7.8 to 7.0 caused an approximately 40-fold decrease in DAU affinity. Overall, we conclude that DAU’s effect on channel activation is due to its direct competition with protons for the pool of high-cooperative sites; the effect on SSD could also be a consequence of the partial occupation of these sites.

To perform a more precise analysis of the DAU-treated channels, the traces for protons activated and protons plus DAU activated averaged currents were subtracted (Figure 8). Area 1 in the plot reflects the delay in the activation of the main peak, which results in an inhibition of the integral current in a very narrow interval during the main peak’s growth. Area 2 is approximately identical to the control current. Area 3 starts at 2.5 ± 0.1 s and reflects the prominent gain of the integral current.

The second peak’s appearance can be explained from various points of view. Based on observations of the two peaks’ similar desensitization time constants, and assuming that DAU does not act on the non-cooperative site, we propose the following hypotheses:

a) DAU binds to the population of the SSD channels (D_SSD_) and converts them from their usual closed state (C) to the additional closed state (C_2_), which cannot be activated as quickly as C can. The activation of the channel from the C_2_ state is characterized by a slow proton-binding rate (i.e., a greater time for each site’s protonation). Because the activation process retains high cooperativity in the proton-binding events, a significant latent period (area 2) develops. Protons consistently fill the proton sites during the latent phase, but the channel remains closed until all the sites have been occupied.

b) DAU influences the channel in the usual desensitized state (D) and converts it into the closed state C, which reopens as a delayed second component during the continuation of the same stimulus. The D→C state transition is stochastic, and it becomes substantially noticeable after most of the channels have been desensitized (starting at 50% channels for t_start_ in Figure 8 and reaching the maximal effect with 85% desensitized channels for t_max_). During one such channel transition, DAU loses the ability to cause further D→C channel state transition; this can occur, for example, via protons displacing DAU from the binding site. 

The second hypothesis was developed to explain the effects that other modulators (such as RF-amide peptides) have on ASIC1a. In this case, a permanently bound ligand shifts the D→C state equilibrium following the many repetitions of the channels’ reactivation process (sustained current).

## 5. Conclusions

pH dependence of rodent ASIC1a activation has a non-uniform shape. This phenomenon can be explained by the model with 9 high-cooperative sites and additional non-cooperative site. DAU has an impact on these high-cooperative sites. The overall effect of DAU application included the simultaneous inhibition and potentiation of the integral current; thus, this ligand can be considered a Janus-faced modulator. Finally, the discovery of DAU’s unusual mode action on the ASIC1a channel revealed the limited nature of the knowledge on this channel’s functioning. We hope that this novel research instrument will be helpful in further studies of ASICs.

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
