# Peer review of "Multiple Modulation of Acid-Sensing Ion Channel 1a by the Alkaloid Daurisoline"

_biomolecules, 2019, doi:10.3390/biom9080336_

Round 1

Reviewer 1 Report

In this report, the authors found the interesting effects of daurisoline (DAU) on ASIC1a channel. They demonstrated that the DAU elicits a second peak component of ASIC1a current, which is different from the sustained component of the ASIC1a. Further, they explored the mechanisms of DAU, which involve an inhibition of ASIC1a activation as well as steady-state desensitization. The experiments were well designed and the manuscript were well written. The discovery provides new insights on physiological mechanisms of ASIC1a channel in response to DAU.

Some minor concerns are listed below.

1. In “Abstract”, Line 28, Daurisoline should be “DAU”.

2. The effect of DAU on hHEG was reported in 2012 (reference 31 in the present submission). The authors should compare the effects of DAU on ASIC1a and hHEG. This should be addressed in “Discussion”.

Author Response

Point 1: In “Abstract”, Line 28, Daurisoline should be “DAU”.

Response 1: The change was done.

Point 2: The effect of DAU on hHEG was reported in 2012 (reference 31 in the present submission). The authors should compare the effects of DAU on ASIC1a and hHEG. This should be addressed in “Discussion”.

 Response 2: We included the comparison of principal effects for DAU onto both channels in the manuscript: “effect DAU onto another known molecular target, hERG channel, is not simple too [35]. DAU shifts the steady-state inactivation of hERG as in the case of ASIC1a. On the other hand, DAU does not affect the activation gating of the hERG channel but it shifts the activation curve of ASIC1a. Finally, DAU produces the reduction of total hERG current via acceleration of the onset of inactivation, while ASIC1a current is enlarged due to the second fast-inactivated component.” (lines 350-355).

We would like to note that this is not the first example of ligands’ cross-reactivity between ASIC and voltage-gated channels. For example, the known peptide ligand of ASIC3 channels, APETx2 toxin, can also effectively inhibit voltage-gated sodium channels (Blanchard et al., Br J. Pharmacol., 2011; Peigneur et al., The FASEB Journal, 2012). Since the spatial structure of the peptide ligand is larger and may include more than one active determinant we decided not to include this speculation about APETx2 toxin in “Discussion”.

Reviewer 2 Report

Summary :

The authors have identified a new and unusual mode of action on the acid-sensing ion channel 1a (ASIC1a) using daurisauline (DAU), an alkaloid from the Asian moonseed Menispermum dahuricum. Preconditioning with DAU at pH7.8 was found upon activation to produce a second prominent component in the micromolar range (EC50 of 23 in oocytes). Interestingly, DAU was found to simultaneously inhibit and potentiate the integral current. From this work the authors propose a new model of proton interaction where in addition to a set of cooperative sites, a non-cooperative site must also be occupied for the channel to transition to the open state. The discovery of a new atypical ASIC modulator - along with a detailed pharmacological analysis of this compound - and the development of a channel behaviour model to integrate the new complexity of the proton interactions will be of interest to the readers of Biomolecules.

-      Manuscript is very well written. There were very few grammatical errors or typos.

-      More information on daurisauline should be presented in the introduction.

-      I agree that the Fig 6 curve fit suggests that in addition to cooperative sites, it is highly likely that a non-cooperative low affinity 10thproton binding site is affecting the ASCI1a channel kinetics. It is appreciated that the development details of this model are included in the Supplementary Materials.

-      The reproduction of the CHO cell data in Fig 6b from Stephan et al. (Fig 1k, ref 34) is helpful and provides reassurance that the channel kinetics observed in oocytes are reproducible in mammalian cells. I did notice one detail however: why are the error bars noticeably smaller for the data points at pH 5.5 and 5 in this manuscript?  

-      I believe that the manuscript would be strengthened by confirming the effect of DAU on ASCI1a in a mammalian expression system, such as CHO cells.

Minor :

-      Please enlarge the smaller type on figures 6 and 7 (legends) and wherever possible on the other figures.

-      Fig 6d: What does the outlying blue data point at pH5 indicate?

-      Line 342: Change ‘shift of the’ to ‘shifts the’

-      Line 347: Insert ‘a’ after ‘appearance of’

Author Response

Point 1:Manuscript is very well written. There were very few grammatical errors or typos

 Response 1: We are grateful for this assessment of our manuscript.

Point 2:More information on daurisauline should be presented in the introduction.

 Response 2: We added additional information to the manuscript (lines 63-66).

Point 3:I agree that the Fig 6 curve fit suggests that in addition to cooperative sites, it is highly likely that a non-cooperative low affinity 10thproton binding site is affecting the ASCI1a channel kinetics. It is appreciated that the development details of this model are included in the Supplementary Materials.

 Response 3: We agree with you that the details of the development of this model are more relevant in the Supplementary Materials section.

Point 4:The reproduction of the CHO cell data in Fig 6b from Stephan et al. (Fig 1k, ref 34) is helpful and provides reassurance that the channel kinetics observed in oocytes are reproducible in mammalian cells. I did notice one detail however: why are the error bars noticeably smaller for the data points at pH 5.5 and 5 in this manuscript?

 Response 4: A stable normalized current amplitude (small error bars) is a result of greater absolute currents induced by a more strong acidic stimulus (pH value 5.5 and below), which led to a saturation effect. We calculated the maximum amplitude (Imax) by individual fitting for each cell. This mathematical operation led to the error bars constriction in a region close to saturation.

Point 5:I believe that the manuscript would be strengthened by confirming the effect of DAU on ASCI1a in a mammalian expression system, such as CHO cells.

 Response 5: We certainly agree with you. However, unfortunately, for reasons beyond our control, a mammalian expression system is not available now.

Point 6:Please enlarge the smaller type on figures 6 and 7 (legends) and wherever possible on the other figures.

 Response 6: We checked all Figures and enhanced legends readability.

Point 7:Fig 6d: What does the outlying blue data point at pH5 indicate?

 Response 7: Sorry, we have not worthy explanation for this experimental fact.

Point 8: Line 342: Change ‘shift of the’ to ‘shifts the’

Point 9: Line 347: Insert ‘a’ after ‘appearance of’

 Response 8 and 9: Thank you for these corrections, the mistakes were fixed.